# Validation of Forest Vegetation Simulator Model Finds Overprediction of Carbon Growth in California

**Claudia Herbert** [1],*, **Jeremy S. Fried** [2] **and Van Butsic** [1]

1   Department of Environmental Science, Policy, and Management, University of California, Berkeley, CA 94720, USA; vanbutsic@berkeley.edu
2   United States Department of Agriculture Forest Service, Pacific Northwest Research Station, Portland, OR 97331, USA; jeremy.s.fried@usda.gov
*   Correspondence: claudiaherbert@berkeley.edu

**Abstract:** Using regression-based, bootstrapped equivalence tests, and remeasured inventory plot data from thousands of plots across California, we found that the Forest Vegetation Simulator (FVS), as typically used out-of-the-box, overpredicts carbon sequestration in live trees that remain alive ten years later by 27%, on average. We found FVS growth prediction sensitive to forest type and FVS variant, with the largest overpredictions occurring in stands within the North Coast variant, growing on the lowest site class, having ages that are unknown or between 50 and 100 years, and that are within governmentally designated reserved areas or on national forests. Direction and magnitude of errors are related to the stand attributes; these relationships point the way towards opportunities to improve the underlying growth models or calibrate the system to improve prediction accuracy. Our findings suggest that forest managers relying on out-of-the-box FVS growth models to forecast carbon sequestration implications of their management of California forests will obtain estimates that overstate the carbon that can be sequestered under light-touch or caretaker management, potentially leading to management decisions that fail to deliver the expected carbon sequestration benefits—a failure that could take a long time to recognize.

**Keywords:** Forest Vegetation Simulator (FVS); model validation; California; equivalence tests; carbon

## 1. Introduction

In response to the threat of climate change, climate policies often highlight forest ecosystems as both a critical carbon sink and at risk from increased environmental stress accompanying climate change [1,2]. Policies for financing sustainable forest management across borders have evolved from programs aimed at reducing deforestation in the global south (e.g., Rainforest Alliance, REDD) to programs crediting forests anywhere for the carbon they store and sequester via carbon markets (e.g., REDD+, California Compliance Market Forest Offset Protocol). The scale of capital interested in reducing global emissions, and the potential for relatively cheap sequestration and storage in forests as offsets, have placed a spotlight on forests as Natural Climate Solutions [3]. In practice, revenue from carbon offsets can pay forest owners for keeping forests intact (avoided conversion), expanding or preventing loss of forests through re/afforestation, or increasing density of forest stocks through improved management [4]. The most common type of carbon offset by number of projects and tons credited in the United States for both the compliance and voluntary markets is this latter approach to offsets known as Improved Forest Management (IFM [5]. IFM offsets assign credit to changes in management (e.g., increasing rotation length, reducing harvest intensity, etc.) that increase carbon sequestered and provide an associated climate benefit. Outside of monetizing carbon for forest management, the largest forest ownership by area and volume in the United States is the network of public forestlands managed by the Forest Service, which are not currently eligible for carbon credits. Despite the absence of carbon sequestration from the current list of objectives

guiding the agency, the Forest Service recognizes the importance of maintaining forests as a net carbon sink; over the past decade, the Forest Service has indicated that increasing carbon sequestration is part of their plan to mitigate climate change [6]. Arguably, the policy interest in forests and the need for accurately characterizing carbon dynamics is now greater than it has ever been.

Developing carbon-focused management plans and participating in a market for carbon credits requires consistent and accurate accounting of forest carbon stores and sequestration into woody biomass. For IFM offsets sold in the California Compliance Market, this is more than USD 1.7 billion worth of forest offsets predicated on the ability of projects to maintain the initial above-average inventory [7]. Despite progress in developing comprehensive forest carbon stock and flux maps, data necessary for the more granular, project-level decision making needed for carbon markets are often limited in geographic and temporal extent [8]. The most accurate practical approach to assessing carbon stocks relies on forest inventory measurements of tree height and diameter. Accurate flux measurement depends on either continuous measurement of canopy gas exchange or taking the difference of two or more stock measurements collected at different times, which is more temporally coarse but also more feasible over large areas. Retrospective analyses based on forest inventory assessments of carbon dynamics have only recently become possible with the availability in the U.S. of remeasured forest inventory data from a spatially balanced sample of all forested lands [9]. To credit current and future carbon sequestration, offset markets rely on periodic forest inventories and selected, peer-reviewed growth and yield models. For example, an IFM offset project can be inventoried as infrequently as once every twelve years. Over that interval, the project can rely on growth and yield models to estimate carbon stocks and flux to document management tradeoffs against anticipated carbon benefits [10]. As forest carbon remains central to climate policy, the accurate performance of these growth and yield models should also be viewed as central to climate policy. Validating the accuracy of growth and yield models is an important step in connecting forest management with an honest characterization of climate benefits to support informed decisions about forests.

The Forest Vegetation Simulator (FVS) is one of the models approved by all carbon offset registries that accept Improved Forest Management [11–14]. The FVS has been developed and supported by the United States Department of Agriculture Forest Service for over three decades, developed to evaluate treatments to help managers make plans that address various forest objectives. This individual tree, distant-independent (not requiring spatial referencing of trees) empirical growth and yield model is widely used to predict stand dynamics, with and without management, in forests covering all ownerships across the United States [15]. The base FVS model predicts growth in tree diameter and height and the probability of mortality over a user-selected period for each tallied tree within a sampled stand. The aggregated projections of tree growth and mortality can be used to estimate future carbon stocks and stock change, as a proxy for carbon flux. Several extensions to the FVS have been developed to account for changes in growth and carbon dynamics that might be expected under climate change, fire disturbance, and infestations by insects and pathogens [16–18]. The full FVS system can be used to evaluate, for example, how forest carbon dynamics and other objectives for the forest would respond to different choices for rotation length, stand density management, and harvest operations to support strategies that maximize credit for forest carbon sequestration [19].

Forest management strategies aimed at maximizing carbon credits may vary by location and context. This results in IFM carbon offset protocols avoiding prescriptive specifications about what management qualifies as "improved". Instead, they offer latitude for demonstrating, via rigorous analysis, both the current trajectory of carbon stocks and how this would change under silviculture designed to enhance climate benefits. Silviculture enhancing climate benefits could include increased retention of previously sequestered carbon or new sequestration, such as storage of additional carbon both in the forest and in harvested wood products and solid waste disposal sites. Some researchers who use the FVS to strategize which types of management can optimize future carbon have re-

ported that limiting forest management through avoiding harvest, lengthening rotations, or maintaining near-maximum retention will result in greater climate benefits via reduced emissions and increased sequestration [20,21]. The FVS was developed for evaluating different management scenarios, and yet carbon-focused management often suggests non-management, most notably on national forests. The validity of FVS-based analyses depends on both analyst assumptions and the FVS generating accurate predictions of forest carbon trajectories. As the FVS is increasingly used to simulate growth without management, our expectations concerning continued (baseline) carbon sequestration are dependent on its predictions.

As resource decisions with long-term consequences for the global climate are increasingly reliant on models like the FVS, the need for robust model validation that exposes model bias and error has never been more important. A recent uptick in publications addressing model validation notwithstanding, fewer than one percent of published modeling studies focus explicitly on validation [22], perhaps owing to limited access to the consistently measured, longitudinal forest data that is essential for the robust evaluation of model performance. The recently published remeasurement data from permanent Forest Inventory Analysis (FIA) plots are ideal for evaluating the validity of FVS growth predictions, as they provide a spatially balanced, statistically representative foundation for assessing FVS performance for multiple variants, forest types, and myriad sizes and species of trees [17].

Previous evaluations of the FVS reported both under- and overpredictions of tree growth, with sign and magnitude varying by model variant, species, and tree size [23–27]. An important motivation for evaluating errors in FVS predictions is to identify conditions where it may be necessary to recalibrate and otherwise improve the model. In some cases, when deviations between predicted and actual growth exceed what is needed for the FVS to be useful, re-engineering of model components may be advisable [28]. The FVS is not one model; each of its 22 variants applies to a non-overlapping, geographically distinct area within the United States, and with multiple component models, parameterizations, and assumptions believed to be appropriate for the species, forest types, and environmental attributes within the area to which the variant applies [17]. Only a few of these have been formally validated against observed tree growth. Notably, we were unable to find any studies evaluating FVS predictions of tree growth in California. This study appears to be the first formal validation of multiple FVS variants using a consistent, spatially balanced sample of longitudinal observations of forest growth, which presents the opportunity to compare prediction errors across variants, as well as species and stand characteristics.

This paper reports the performance of the FVS when used to project forest carbon stocks and flux. This is an important area of research given the increasing use of FVS as a carbon calculator in support of carbon market offset projects. We evaluated the four FVS variants that account for 99% of California's forest land and for which no FVS validation exists. Specifically, we sought to address the following questions:

1. Are 10-year FVS projections of carbon stocks equivalent to field measured carbon stocks?
2. Are FVS predictions of growth in carbon equivalent to observed growth over 10 years?
3. If stocks or growth are not equivalent,
   a. Which types of forest lands generate predictions with the greatest departures from observed values?
   b. Which species and tree sizes lead to predictions with the greatest departures from observed values?

## 2. Materials and Methods

### 2.1. Study Area Description

We analyzed data from 3340 FIA plots that met criteria for inclusion (plots for this analysis sampled forest land that remained forest land, contained live trees at the initial visit, and were neither treated nor disturbed by fire), downloaded from the public Forest Inventory and Analysis Database (FIADB) repository (version 7.0, California file, part of

the Pacific Northwest Work Unit, at https://apps.fs.usda.gov/fia/datamart/datamart.html (accessed on 10 February 2023)) representing nearly all forested lands in California (Figure 1). These forests span a broad range of productivity classes and forest types, from redwood to California mixed conifer, and include dry forests dominated by oak and pinyon–juniper, where carbon stocks, and growth, are typically much lower. Based on the FIA measurements at the initial visit (2001–2009), when these plots were installed under the annual inventory design, which samples the forest over a ten year cycle via annual panels of plots that are subsequently remeasured in the next ten-year cycle, most forest carbon can be found on Federal lands, except in the Klamath Mountains (NC) variant, where private forests hold more carbon (Table 1). The largest carbon stocks tend to be on forests in the 50–100-year age class, perhaps because these account for the plurality of forest area, except for in the Western Sierra Nevada (WS) variant, where carbon stocks are greatest in 100–150-year-old forests. Including the entire state in this analysis offered the opportunity to evaluate the performance of multiple FVS variants across diverse forest landscapes.

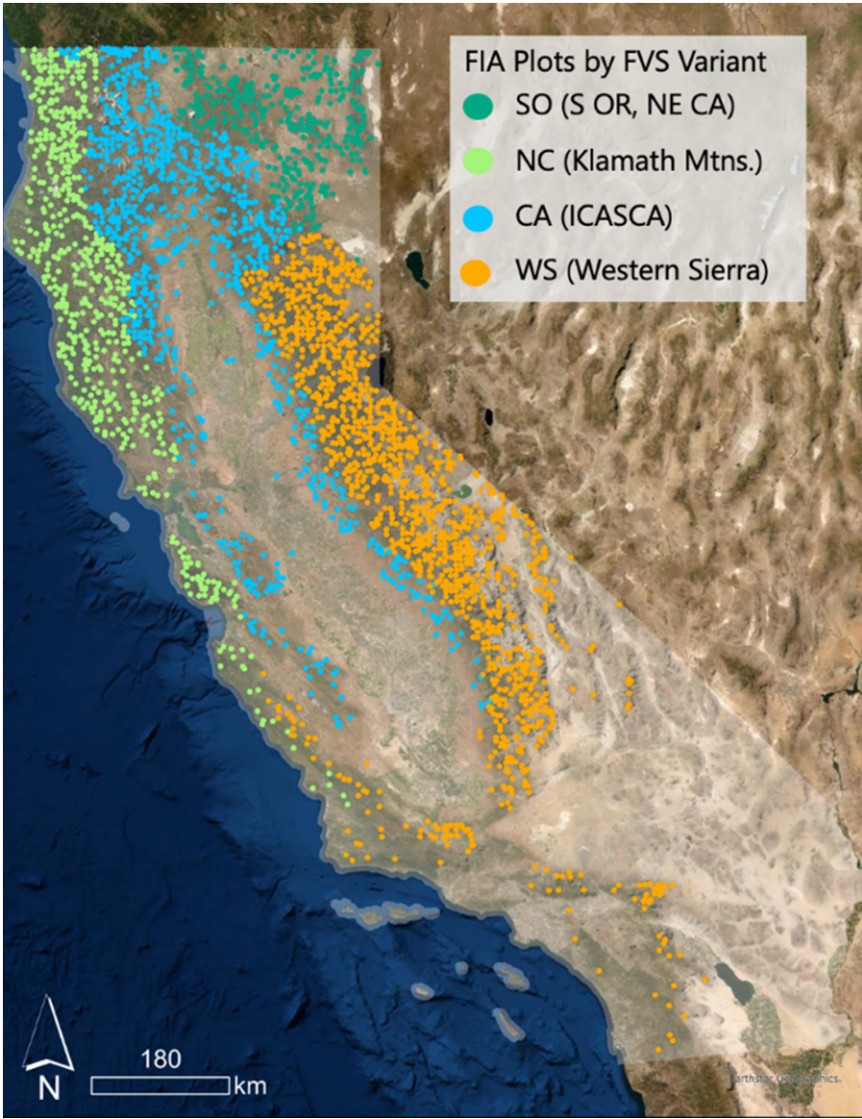

**Figure 1.** Approximate locations of FIA plots in California used for this analysis, shaded by FVS Variant: SO = South Central Oregon and Northwest California, NC = Klamath Mountains, CA = Interior California and Southern Cascades, WS = Western Sierra.

**Table 1.** Live tree carbon stocks observed at the remeasurement visit and flux calculated as stock change between the installation and remeasurement visits, computed using FIA's system of allometric models for tree carbon for trees observed, at both visits, to be live and growing on land classified as forest, by stand age class and variant, and proportions of carbon stocks by owner group (dark green is national forest, light blue is other public, and grey is private).

| Summary Carbon Stock by Variant and Stand Age | | | | |
|---|---|---|---|---|
| **FVS Variant** | **Ownership Proportion** | **Stand Age (Years)** | **n** | **Aboveground Live Carbon (Mg/Hectare)** |
| Inland California and Southern Cascades (CA) | | $\leq$50 | 88 | 825,569 |
| | | 50–100 | 315 | 6,265,330 |
| | | 100–150 | 171 | 4,752,099 |
| | | 150–200 | 52 | 1,541,091 |
| | | >200 | 58 | 2,485,565 |
| | | Multi-age or unknown | 185 | 1,551,650 |
| Klamath Mountains (NC) | | $\leq$50 | 154 | 3,573,573 |
| | | 50–100 | 202 | 6,807,758 |
| | | 100–150 | 88 | 4,291,585 |
| | | 150–200 | 38 | 2,568,082 |
| | | >200 | 38 | 2,429,303 |
| | | Multi-age or unknown | 85 | 1,385,899 |
| South Central Oregon and Northeast California (SO) | | $\leq$50 | 29 | 132,856 |
| | | 50–100 | 187 | 1,886,297 |
| | | 100–150 | 90 | 1,738,521 |
| | | 150–200 | 18 | 337,833 |
| | | >200 | 23 | 566,941 |
| | | Multi-age or unknown | 4 | 7,547 |
| Western Sierra Nevada (WS) Private / Other Public / Federal | | $\leq$50 | 108 | 805,302 |
| | | 50–100 | 327 | 7,701,272 |
| | | 100–150 | 275 | 8,853,684 |
| | | 150–200 | 160 | 4,932,200 |
| | | >200 | 201 | 6,252,942 |
| | | Multi-age or unknown | 230 | 1,603,264 |
| | | Total | 3126 | 73,296,163 |

## 2.2. The Forest Inventory and Analysis Database Description

To model forest growth in the FVS, and to discover the drivers of model validity, we relied on data from three tables in the FIA database (The Forest Inventory and Analysis Database: Database Description and User Guide for Phase 2 (version 9.0.1)): Plot (for location attributes such as elevation and applicable FVS variant), Condition (for site attributes such as ownership, forest type, site class, and evidence of disturbance and management), and Tree (species, diameter, and height) [29]. Most of these data are collected in the field by experienced forest inventory crews operating within a rigorous quality assurance framework; computed attributes (e.g., tree volume and carbon) are calculated as part of the FIA compilation in the months following completion of an annual data collection panel.

The FIA's average sampling intensity in California is a nominal one plot per 2387.6 hectares. Each plot consists of three component plot sizes, the area of which is distributed equally around four sample points: four 2 m radius microplots, totaling 1/75th acre, to sample trees 2.5–12.6 cm dbh; four 7.3 m radius subplots, totaling 1/6th acre, to sample trees between 12.7 cm and 60.96 cm; and four 17.9 m radius macroplots, totaling 1 acre, to sample trees larger than 60.96 cm in diameter (Supplemental Figure S1). On plots

containing more than one ownership class, land use, forest type, reserve status, stand size class, or stand density class, these differences are mapped in the field, subject to minimuim size and width requirements, as distinct "conditions", each with its own set of condition attributes and a "condition proportion" that accounts for the mapped area of the condition relative to the full plot footprint. Plots without such condition mapping have a single condition.

We modeled each FIA condition as mapped at the installation visit (Time 1) in the FVS as a "stand", and all the trees projected by the FVS on that condition, or observed at the remeasurement, were assigned the site context values observed on the condition at the Time 1 visit. Carbon stocks and carbon growth of all trees on a condition are summed, weighted by the inverse of the plot size on which they were sampled and adjusted for the condition proportion observed at Time 1 to arrive at carbon stock and flux due to growth for the condition. We refer to FIA conditions as stands, consistent with FVS terminology, for the remainder of this paper. To examine the role of tree size and species on FVS prediction performance, we also performed a tree-level validation of carbon stock and flux.

Under the enhanced annual inventory system [30] that began in California in 2001, FIA plots in the western US are visited and assessed at ten-year intervals, with ten percent of the plots (one panel) receiving a visit each year. This analysis includes all FIA plots in the annual inventory system that were first assessed in one of the nine panels assessed in 2001–2009 and remeasured approximately ten years later (remeasurement intervals for 9 percent of plots depart from the intended 10 years, typically by not more than 1–2 years, owing to delays presented by logistical challenges such as denied access and wildfire closures).

We sought to test the FVS's capacity to predict forest growth in the absence of management and large-impact disturbance because, while those influences can be modeled in the FVS and the performance of those model functions is of interest, the specific form of those disruptions to growth (e.g., fire severity, kind of thinning regime) could not be determined from the FIA data with sufficient precision to parametrize in FVS simulations. Stands for which crews implementing the remeasurement visit coded fire disturbance or any kind of human-initiated tree removal or surface disturbance (e.g., prescribed fire, chaining) during the remeasurement period were dropped, as were stands that changed status from forest to non-forest (conversion) or vice versa (reversion) between inventory visits, based on codes in FIADB's TREE_GRM_MIDPT table. These filtering rules removed 709 stands, mostly owing to fire disturbance or management activity. Where stands were coded as having ≥25% of the trees or sample area affected by insects, disease, weather damage, or geologic disturbances damage between inventory visits, we coded a binary, non-fire disturbance attribute to have available as an explanatory factor but did not drop them from the analysis set. Stands that had any type of observable treatment during the remeasurement period were removed to ensure that comparisons with our "grow-only" FVS model runs would include only stands that did, in fact, grow-only. The ten percent of live trees that, for a variety of reasons (e.g., forking, swelling, an active wasp nest), had diameter measured at other than breast height at one or both visits, as indicated by the "Diameter Check" code in the Tree table, were removed because diameter is a critical input for FVS growth simulations and the FVS is not designed to adjust diameters collected at non-standard heights (Figure 2). These trees were removed only after the FVS modeling so that their presence as competitors for resources would be reflected in modeled stand growth. The filtered dataset contains 69,480 trees on 3024 stands representing 42 different forest types and located as all or portions of 2920 plots.

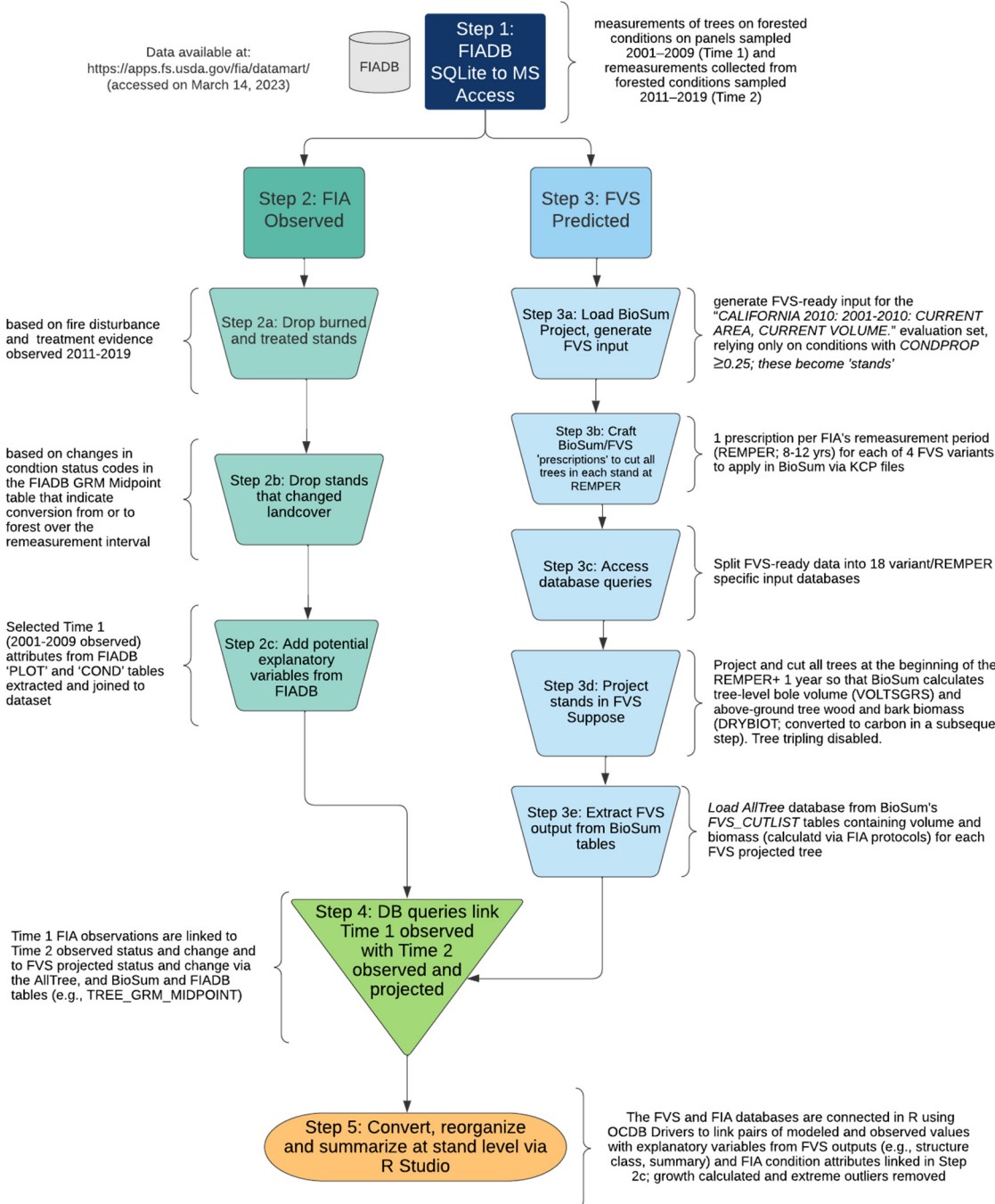

**Figure 2.** Data and workflow diagram for validation analysis.

### 2.3. Modeling Tree Growth with the Forest Vegetation Simulator

FIA data for the 2001–2009 panels were prepared for modeling in the FVS by loading them into the Bioregional Inventory Originated Simulation Under Management (BioSum) version 5.87 software, which assigned the correct FVS variant for each plot and generated FVS input files for each variant that could be used to perform a grow-only simulation for the remeasurement period (Figure 1) [31]. There were 1301 stands in the WS variant, 867 stands in the Inland California and Southern Cascades (CA) variant, 603 stands in the Klamath Mountains (NC) variant, and 347 in the South Central Oregon and Northeast California (SO) variant. We did not model the 16 stands in the Central Rockies (CR) variant because that variant contains <1% of eligible stands in California and any conclusions reached based on such a small sample would not be generalizable to the variant, which covers many

states. The four variants (WS, CA, NC, and SO) were combined with what turned out to be the actual (as opposed to targeted 10-year) remeasurement periods (from as short as 8 to as long as 12 years) to create 18 variant/remeasurement period combinations that were simulated separately with appropriate specifications for the variant and growth period.

To evaluate FVS performance in estimating volume and carbon growth in live trees that remained alive, and volume and carbon stocks at the end of the growth period, we initiated an FVS simulation in the Suppose user interface (version 2.08) with no forest management activities—what we call a "grow-only run". We supplied all required and virtually all optional FVS stand and tree-level inputs (e.g., diameter, height, crown ratio, and, via computations within BioSum, FVS compatible site index) (Figure 2). We avoided any customization or adjustments. Although some users have developed custom adjustments (e.g., overrides to the maximum allowable stand density index for a species via FVS's SDIMAX keyword) to achieve more realistic projections, we chose not to use these, consistent with our goal of evaluating the performance of the FVS "out-of-the-box". We also did not provide the previous diameters or heights that FVS can use as a basis for model calibration, because these were not consistently available in our dataset, which comprised FIA initial installation visit data, and are not typically available for most users. The simulations projected tree growth over the remeasurement period, then "cut" all trees so that when "cutlists" were subsequently loaded into BioSum, volume and biomass would be automatically calculated via FIA-curated equations embedded in BioSum. This ensured that carbon estimates for both projected and observed growth in tree diameter and height would be calculated via the same system of equations. The purpose of this validation was to compare the policy-relevant predictions (i.e., of volume and carbon) associated with modeled diameter and height growth, not the impact of volume and biomass allometry choices by the FIA and FVS programs.

*2.4. Tree-Level Carbon and Volume Stocks and Growth*

To evaluate nominally 10-yr predictions of carbon stocks, we compared live tree, non-foliar, above-ground carbon, calculated as half of DRYBIOT (total above-ground tree biomass, oven-dry weight), computed using PNW-FIA's standard equation system, for tree diameters and heights projected from Time 1 (the installation visit) by FVS to the measured diameters and heights of those trees at Time 2 (the FIA remeasurement visit). Volume growth predictions were based on PNW-FIA's VOLTSGRS attribute (gross volume total stem, which includes all bole wood from ground to tip). We expected that projected stocks would not differ greatly from observed stocks for such a short (~10 years) projection period because for stands free of disturbance and treatment, the predicted growth increment would typically be small relative to stocks, and most of the variation in Time 2 stocks would track variation in Time 1 stocks. Of greater interest is the performance of the FVS in predicting carbon accumulation, in other words, the rate at which carbon becomes sequestered in live trees. We calculated this, for volume and carbon, as "Annual Growth Increment" (Equation (1)).

$$Annual\ Growth\ Increment \left( \frac{Mg/ha}{yr} \right) = \frac{[FIA\ Time\ 2\ or\ FVS\ Modeled\ -\ FIA\ Time\ 1]\,(Mg/ha)}{Remeasurement\ period\ length\,(yr)} \quad (1)$$

Regional calculations of biomass for each tree in the PNW FIA unit derive bole (stem) wood biomass from cubic foot volume (VOLTSGRS) and species-specific wood density [32]; branch and bark biomass are calculated from equations developed for that tree's species or a related one [33,34]. Carbon and volume are closely related because bole wood, the basis for volume calculation, accounts for about 80 percent of the carbon in the above-ground portions of a tree, on average, and 65 percent when also considering the belowground portions. All units not in Système International d'unités (SI) were converted before analyzing results using the "measurements" package in R [35].

$$tree\ bole\ wood\ carbon\ (Mg) = \left( \frac{\left( Volume(ft3) \times wood\ density \left( \frac{lbs}{ft3} \right) \right)}{2000 \left( \frac{pounds}{short\ ton} \right)} \right) / 0.9071 \left( \frac{short\ ton}{long\ ton} \right) \tag{2}$$

### 2.5. Stand-Level Carbon and Volume

FVS projected and FIA Time 2 observations of tree volumes and carbon were aggregated to stand (FIA condition) level as weighted sums using the ratio of trees per acre unadjusted (TPA_CURR_UNADJ) and condition proportion (CONDPROP) from Time 1 as the weight (Equation (3)). TPA_CURR_UNADJ is the inverse of the plot size on which a tree was sampled; for example, a tree between 12.7 cm and 60.96 cm in diameter at Time 1 was sampled on a 1/6-acre subplot, so it would have TPA_CURR_UNADJ = 6. However, if that tree existed on a condition covering only half of the plot (CONDPROP = 0.5), then for the purpose of carbon density calculation on a per-acre basis that sample tree represents 12 trees. Volume and carbon were calculated as a density (i.e., expressed on a per hectare basis).

$$Stand\ stock\ or\ growth = \sum_{all\ trees} tree\ stock\ or\ growth \times \frac{TPA\_CURR\_UNADJ}{CONDPROP} \tag{3}$$

For both Annual Growth Increment and Time 2, we removed extreme outliers (observations that exceeded the 3rd quartile plus three times the interquartile range [IQR]) from the remainder of the analysis as they do not appear to be linked to any aspect of FVS projection. Excluding these extreme outliers removed less than 3% of observations, for a total of 3118 trees that were outliers due to either the FIA observation or the FVS prediction.

### 2.6. Validation through Evaluation of Bias and Accuracy

The first validation diagnostics are numerical descriptions of bias and accuracy of carbon and volume growth and stock values. Bias, or the systematic error, is the average deviation of repeated estimates from the true value (Equation (4)) [36]. Bias shows the direction of systematic model deviation: in this context, negative signs for bias mean that FVS overpredicts the growth observed on FIA plots. The RMSE is also scale-dependent but will always be positive—it is a measure of accuracy, sensitive to outliers, where higher RMSE values indicate lower model accuracy (Equation (5)). Bias and RMSE are the first steps in understanding FVS model accuracy relative to the FIA observations and, in the case of bias, the direction of model deviation. Bias and RMSE are reported for the stand-level carbon and volume Time 2 stocks, the corresponding annual growth increments, and a relative growth measure, defined here as percent of initial (Equation (6)).

$$Bias = \sum_{i=1}^{n} (actual_i - predicted_i)/n \tag{4}$$

$$RMSE = \sqrt{\sum_{i=1}^{n} \frac{(actual(FIA)_i - predicted(FVS)_i)^2}{n}} \tag{5}$$

$$Percent\ of\ Initial = \frac{time\ 2\ (predicted\ or\ observed)}{time\ 1\ measurement} \times 100 \tag{6}$$

### 2.7. Equivalence Tests

Equivalence testing has been demonstrated as a robust method for evaluating model outputs [37,38]. This approach reverses the conventional null hypothesis assumption that corresponding sets of modeled and observed data are similar and assumes, instead, that they are distinct, thereby shifting the burden of proof onto the modeled data [37]. To test this hypothesis of dissimilarity, a regression-based, two one-sided equivalence test

(TOST) compared the observed data with the mean-shifted FVS modeled data. This TOST is repeated twice: once to compare intercepts and again to compare the slope against a 1:1 slope. The intercept TOST tests for agreement between the mean observed and predicted value. The slope TOST tests for agreement between the plot of modeled and observed data and a 1:1 slope. A slope from the fitting of the modeled to observed data that is near 1 indicates high model accuracy across the range of observations and suggests validation of model structure—or that the model is accurate for the "right" reasons [28,37].

**H$_0$.** *The observations are dissimilar; FVS is not predicting FIA-observed values within the defined Smallest Effect Size of Interest (SESOI) (e.g., +/−10%) of FIA observations for slope or intercept.*

**H$_1$.** *The observations are similar, FVS is predicting FIA-observed values within the SESOI of FIA observations.*

The TOST evaluates equivalence by treating model underprediction or overprediction as two independent tests; model predictions failing on just one side of the test means the model is not equivalent. The Smallest Effect Size of Interest (SESOI) sets the "cutoff" level of equivalence expressed as a percentage deviation from the mean observed value. This SESOI value is selected before conducting a TOST and can be interpreted as the maximum acceptable difference between observed and predicted values before the model would be considered not capable of offering reliable predictions for the purposes for which it is used. We conducted three TOSTs, with equivalence thresholds of 5%, 10%, and 25% that are consistent with previous validations of FVS predictions of tree growth [23,28,37]. We performed a non-parametric bootstrap with 1000 replications to develop confidence intervals around our observation means to see if the predictions fall within our region of equivalence using the "equivalence" package in R Studio using an alpha level of 0.0125 [38–40]. If our SESOI-constructed Equivalence Levels do not fall within the bootstrap-constructed Confidence Levels, then the null hypothesis of dissimilarity is supported. A TOST rejection of the null hypothesis of dissimilarity for slope or intercept at either equivalence level can be interpreted as an equivalence between FVS predictions and FIA observations at that equivalence level. The null hypothesis of dissimilarity should be rejected for both slope and intercept for a model to be considered equivalent.

### 2.8. Using Stand-Level Models to Diagnose Systemic Forest Vegetation Simulator Prediction Errors

Because land management style and vegetation type may account for patterns of over-or underprediction, investigation of these patterns could highlight opportunities for model refinement. We identified five condition-level variables that might drive FVS errors: (i) FVS variant, (ii) site class, (iii) stand age, (iv) reserve status, and (v) land ownership. We transformed stand age as a categorical variable by classification into 50-year bins.

To evaluate the influence of these five variables on model error, we fit a multiple linear regression model. This regression predicts carbon growth increment using potentially explanatory stand-level variables and interaction terms for whether the data point is observed or modeled. Adding interaction terms allow us to fit a linear model with different effects for the FVS and FIA values. This helps tease out the differences for how different values within categories, such as stand age or site class, influence FVS errors in predicting carbon growth increment. When the FVS value exceeds the FIA value, overprediction is indicated; underprediction is indicated when the reverse is true.

The dataset for regression analysis contains two entries for each stand: one with FIA-observed growth as the dependent variable and one with FVS-modeled growth, with all explanatory variables identical and a binary, "run" variable indicating whether the record's dependent variable is FIA-observed or FVS-modeled. The linear model controls for differences in explanatory variables such as stand density, site conditions, and initial stocks of forest carbon to predict the FVS-modeled or FIA-observed growth. Along with the run variable, FIA Time 1 value, and the five variables used as interaction terms, several other variables, drawn from both the FIA database and FVS computed values from the initial visit,

were included: Structure Class, Aspect, Slope, Basal Area, Stand Density Index, Number of Strata, Total Cover, Quadratic Mean Diameter (QMD), Stratum 1 DBH, Forest Type, and Other Disturbance, a variable noting if there was a non-fire disturbance impacting over 25% of the trees or sample area during the remeasurement interval.

Using a Stepwise algorithm based on AIC from the "stats" base package in R, we dropped variables identified as reducing model fit if they were not also a component of an interaction term variable of interest [39]. Because the initial set of variables included some that may be largely redundant (e.g., highly correlated metrics of density which are often collinear), we also used variance inflation factor (VIF) to identify and further remove density and geophysical variables with VIF > 10. The final list of variables included the run variable, FIA Time 1, the five interaction terms (e.g., run by site class, variant, stand age, reserve status, and land ownership), Structure Class, Aspect, Slope, Total Cover, QMD, Stratum 1 DBH, and Forest Type. Site class was one variable that regularly emerged as having a high VIF, suggesting unsurprisingly high collinearity with other variables. For example, productivity is not independent of ownership. We did not drop site class as a predictor variable, despite such collinearity, because we believed it important to explore patterns of FVS error across an explicitly defined productivity gradient.

The regression is implemented using the "lm" function in the R, also in the "stats" package, and model fit diagnostics are done using the R package "olsrr" [39,41]. Data were organized for analysis using the "tidyverse" package and plots were made using "ggplot2" packages in R [42,43]. Because the interaction terms and the number of covariates make it difficult to interpret the coefficients in the linear model output, we used a prediction function to graph the mean predicted values for each category of interest. The output linear model was then used to predict and compare growth values for the FVS versus FIA using the "emmip" function in the "emmeans" package in R [44]. The final linear model was significant at a 0.01 cutoff with an adjusted R-squared of 0.67.

### 2.9. Tree-Level Evaluation: How Does Error Vary by Tree Diameter and Species?

To further evaluate the trends of where bias was observed at the stand level, we used the tree-level measurements to test whether some species or diameter ranges might tend to have greater errors when predicting growth. This information could help prioritize opportunities to improve model predictions. For example, the growth model parameters for a particular species might need updating.

We expected that tree diameter class and species might be important predictors of FVS performance because the FVS relies on tree size relative to the rest of the trees in a stand to calculate growth and allocate mortality, and because for some less common species, growth equations from other species are applied, potentially opening opportunities for less accurate growth predictions for certain species [17]. We created five diameter classes to subset the trees: (i) 2.5 to 12.7 cm, (ii) 12.7 to 25.4 cm, (iii) 25.4 to 53.34 cm, (iv) 53.34 to 76.2 cm, and (v) greater than 76.2 cm. To analyze the role of species in tree-level errors, we selected the top six species by both sample tree frequency—*Pseudotsuga menziesii* (Douglas-fir), *Abies concolor* (white fir), *Pinus ponderosa* (ponderosa pine), *Notholithocarpus densiflorus* (tanoak), *Quercus chrysolepis* (canyon live oak), and *Calocedrus decurrens* (incense-cedar), and by tree volume, which added *Sequoia sempervirens* (redwood), *Abies magnifica* (California red fir), and *Pinus jeffreyi* (Jeffery pine) for a total of nine species. These subsets (for tree size and species) formed the basis of additional equivalence tests, replicating the analytic approach already described for stand level analysis. Subsets where the null hypothesis of dissimilarity cannot be rejected provide an indication of species and sizes for which out-of-the-box FVS simulation of carbon growth may not be valid.

### 3. Results

### 3.1. Bias and RMSE

Considering all forests in California, FVS predictions of carbon flux in live trees that remained alive ~10 years later were 0.05 Mg/ha/year greater than observed (Table 2). The

10-year FVS does not produce projections of carbon flux equivalent to field measured carbon flux. In a state with 12.9 million hectares of forest, this implies a 6.45 Tg overestimation of carbon flux into live trees over the ten-year remeasurement period. The negative signs for bias for both volume and carbon measurements point to FVS overprediction relative to observed growth. Because carbon and volume are recorded in different units and measure different entities (i.e., above-ground live wood and bark versus only bole wood), we calculated relative growth to determine if bias and RMSE are similar between volume and carbon. The rest of this paper presents the findings only for carbon because the relative growth metric showed that relative growth of volume was very close to that of carbon in both bias and RMSE.

**Table 2.** Bias and RMSE results for stand-level above-ground, non-foliar, tree carbon (wood and bark) and bole wood volume, comparing Time 2 (stocks predicted ~10 years forward), Relative Growth (relative growth as a percent of initial stocks), and Annual Growth Increment.

| | Bias Statistics | | | | | |
| | Carbon | | | Volume | | |
| **Measure** | **Time 2 (Mg/ha)** | **Relative Growth (%)** | **Annual Growth Increment (Mg/ha/yr)** | **Time 2 (m³/ha)** | **Relative Growth (%)** | **Annual Growth Increment (m³/ha/yr)** |
|---|---|---|---|---|---|---|
| Bias | −0.50 | −27.20 | −0.05 | −1.58 | −27.73 | −5.63 |
| RMSE | 1.49 | 241.76 | 0.15 | 5.42 | 245.32 | 19.25 |

### 3.2. Equivalence Tests

The TOST intercept results show that the FVS predicted growth increments are not within 10% of the FIA observations at the mean, and the slope result indicates a lack of agreement across stand-level predictions with measured values (Table 3 and Figure 3). The FVS predictions of growth in carbon are not equivalent to observed growth over 10 years. FVS annual growth predictions are equivalent at the 25% level for mean values (intercept test) but not across the range of stand-level predictions (slope test). The predicted carbon stocks pass equivalence tests at the 5%, 10%, and 25% equivalence levels, so FVS projections of carbon stocks, one decade into the future, appear to be valid at the testing thresholds we assumed. For the growth TOSTs that fail to reject the null hypothesis of dissimilarity, we find the proportion of bootstrapped Confidence Levels falling below the Equivalence Level. The larger proportion of the bootstrapped Confidence Level being below the Equivalence Level is consistent with the negative bias results discussed previously, and it confirms that the FVS overpredicts growth rate and stocks relative to observations recorded on FIA plots. As a robustness check, to investigate whether low productivity (<20 ft³/ac/year growth) forests may account for at least some of the underprediction, we repeated the equivalence tests after removing these stands (those with site class = 7). After this change, the Annual Growth Increment passed the slope test, but only at the 25% equivalence level.

**Table 3.** Equivalence test outcomes for intercept and slope for Time 2 (stand-level above-ground, live-tree, non-foliar carbon stocks) predicted ~10 years forward and Annual Growth in those stocks at 5, 10, and 25% smallest effect size of interest (SESOI). Blackened boxes indicate tests that failed to reject the null hypothesis of dissimilarity.

| | | | | Result of Carbon Intercept Test | | | | | | |
| *SESOI* | *Metric* | *Mean Observed* | *Mean Predicted* | *Hypothesis* | *CI Lower* | *CI Upper* | *EI Lower* | *EI Upper* | *Bootstrap Below* | *Bootstrap Within* | *Bootstrap Above* |
|---|---|---|---|---|---|---|---|---|---|---|---|
| 5% | Time 2 | 12.26 | 12.76 | Reject | 12.2 | 12.31 | 12.12 | 13.4 | 0% | 100% | 0% |
| | Annual Growth Increment | 0.21 | 0.26 | Not Reject | 0.2 | 0.21 | 0.25 | 0.27 | 100% | 0% | 0% |

**Table 3.** *Cont.*

| | | | | Result of Carbon Intercept Test | | | | | | | |
|---|---|---|---|---|---|---|---|---|---|---|---|
| *SESOI* | *Metric* | *Mean Observed* | *Mean Predicted* | *Hypothesis* | *CI Lower* | *CI Upper* | *EI Lower* | *EI Upper* | *Bootstrap Below* | *Bootstrap Within* | *Bootstrap Above* |
| 10% | Time 2 | 12.26 | 12.76 | Reject | 12.2 | 12.31 | 11.48 | 14.04 | 0% | 100% | 0% |
| | Annual Growth Increment | 0.21 | 0.26 | Not Reject | 0.2 | 0.21 | 0.23 | 0.28 | 100% | 0% | 0% |
| 25% | Time 2 | 12.26 | 12.76 | Reject | 12.2 | 12.31 | 9.57 | 15.95 | 0% | 100% | 0% |
| | Annual Growth Increment | 0.21 | 0.26 | Reject | 0.2 | 0.21 | 0.19 | 0.32 | 0% | 100% | 0% |

| | | | | Result of Carbon Slope Test | | | | | | |
|---|---|---|---|---|---|---|---|---|---|---|
| *SESOI* | *Metric* | *Hypothesis* | *SESOI* | *CI Lower* | *CI Upper* | *EI Lower* | *EI Upper* | *Bootstrap Below* | *Bootstrap Within* | *Bootstrap Above* |
| 5% | Time 2 | Reject | 5% | 0.98 | 0.99 | 0.95 | 1.05 | 0% | 100% | 0% |
| | Annual Growth Increment | Not Reject | 5% | 0.71 | 0.77 | 0.95 | 1.05 | 100% | 0% | 0% |
| 10% | Time 2 | Reject | 10% | 0.98 | 0.99 | 0.9 | 1.1 | 0% | 100% | 0% |
| | Annual Growth Increment | Not Reject | 10% | 0.71 | 0.77 | 0.9 | 1.1 | 100% | 0% | 0% |
| 25% | Time 2 | Reject | 25% | 0.98 | 0.99 | 0.75 | 1.25 | 0% | 100% | 0% |
| | Annual Growth Increment | Not Reject | 25% | 0.71 | 0.77 | 0.75 | 1.25 | 76% | 24% | 0% |

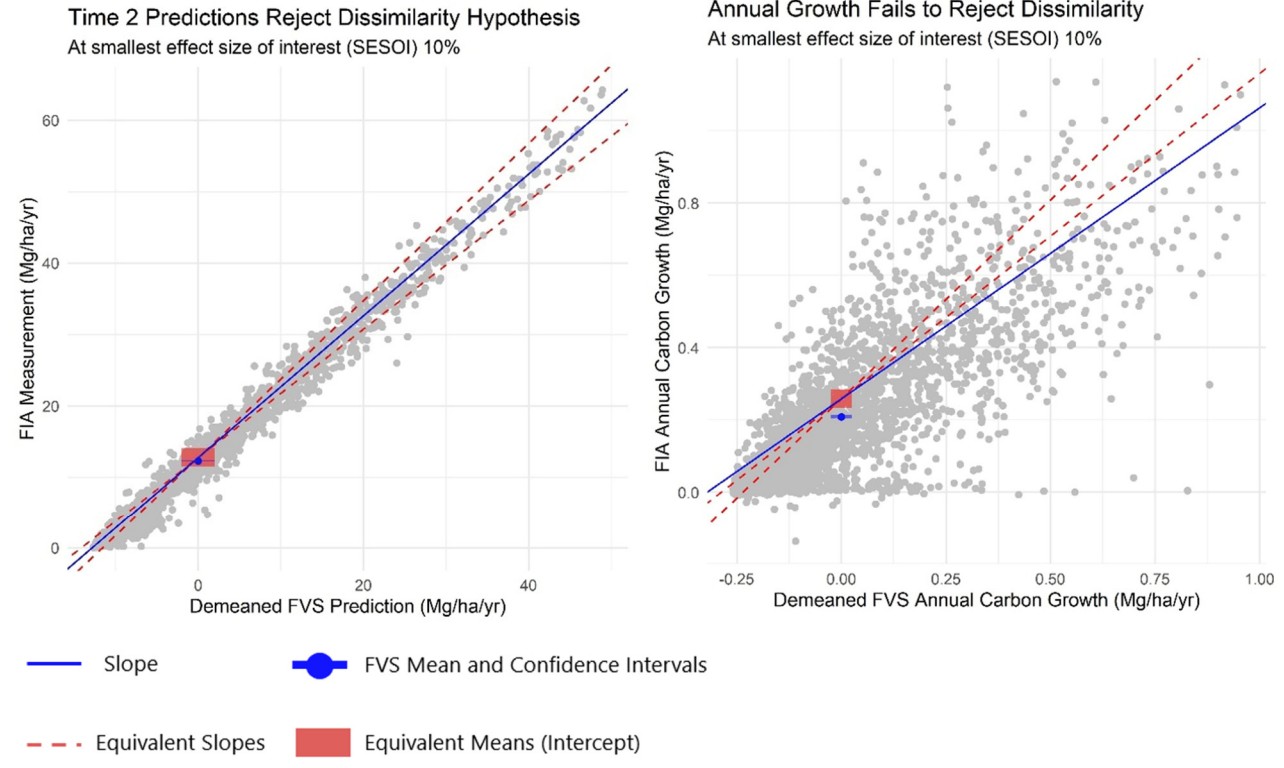

**Figure 3.** Scatter plots of above-ground, live tree, wood and bark carbon at Time 2 (stocks ~10 years following initial measurement) and Annualized Growth for stands with equivalence confidence intervals with smallest effect size of interest (SESOI) at 10%.

### 3.3. Stand-Level Models

Consistent with the FVS overprediction established by the Bias and Equivalence Tests, the linear model predicts a mean FVS value, 0.259, that is 55% greater than the mean FIA value, 0.162, in Mg of carbon sequestered per hectare per year (Figure 4). For the explanatory variables tested with interaction, linear model-output prediction enables assessment of the FVS effect across those variables. Comparing the interaction term levels across the FVS variant, site productivity, stand age, reserve status, and ownership, we

find instances of FVS both over-and under-predicting growth. The differences between the mean FVS- and FIA-predicted increments on these interaction plots are the differences between FVS outputs and FIA observations, holding all other variables constant.

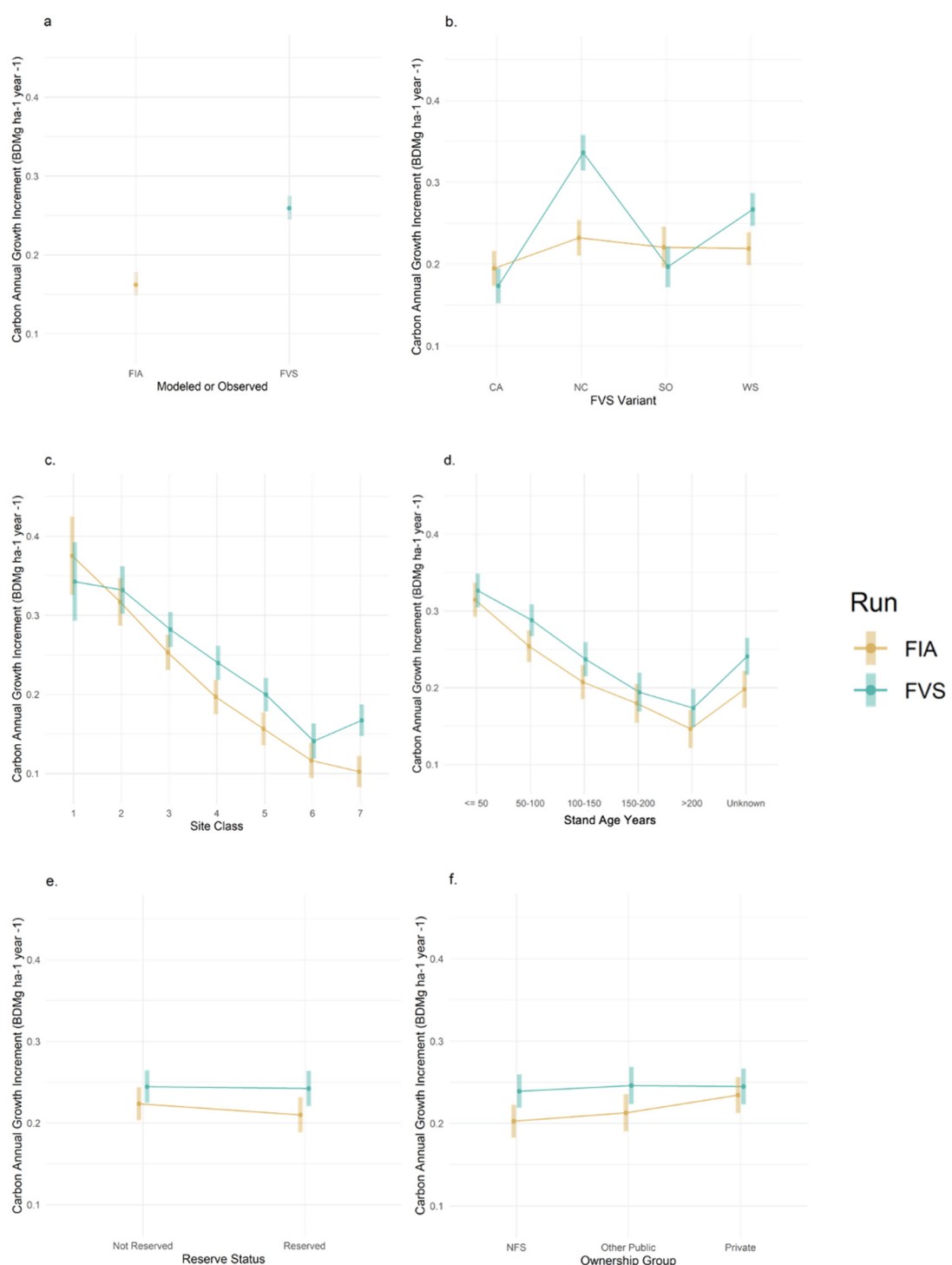

**Figure 4.** Average effect for the run (FIA vs. FVS) variable (**a**) and interaction terms (**b**–**f**), showing the relative effect of each level of each interaction term on the modeled FIA and FVS means for those levels, illustrating where FVS predictions over- and under-state modeled FIA values. FVS Variants (**b**) include Southern Cascades (CA), Klamath Mountains (NC), South Central Oregon and Northeast California (SO), and Western Sierra Nevada (WS).

### 3.3.1. Variant

FVS underpredicts growth for the Southern Cascades (CA) and South Central Oregon and Northeast California (SO) variants by 10% of the FIA predicted values, and overpredicts by 22% of the FIA values for the Western Sierra Nevada (WS) and 44% for the Klamath Mountains (NC) variants. Most of California's forest carbon stocks are within the NC and WS variants (Table 1). Given the modest degree of underpredictions in CA and SO relative to the over predictions, particularly for the NC variant, FVS predicted carbon accumulation rates are most inaccurate for the regions with greatest forest carbon stocks.

### 3.3.2. Site Productivity

For Site Class values of 2 through 7, the FVS overpredicts. FVS error increases as site productivity decreases, (and site class, an ordinal variable that increases as annual volume productivity decreases, increases), with the error greatest for site class 7 where the difference is statistically significant and the FVS overpredicts by 60% of the predicted FIA value. For the most productive lands (site class = 1), the FVS understates stocks and growth, though their overlapping standard error bars suggest this difference is not statistically significant. The progressive decrease in FVS prediction accuracy as site productivity decreases suggests an opportunity to build in a calibration or correction for FVS predictions.

### 3.3.3. Stand Age

FVS overpredicts growth for all age cohorts, with overprediction less (<4%) for the under 50-years cohort and greatest for the 50–100- and 100–150-year-old cohorts. Older cohorts may be dominated by stands that have experienced little or no active management. The FVS was developed to support decisions about management by projecting the results of alternative management activities that are more common in young-growth stands. Older stands under "caretaker" grow-only management may be less well represented in the data used to fit the FVS model. The stands that did not have an age recorded (categorized as age cohort "NA") exhibit FVS over-prediction similar to that for the 50 to 100- and 100–150-year-old cohorts.

### 3.3.4. Reserve Status

The FVS overstates growth for both unreserved and reserved forests, though overprediction is greater for reserved forests (15%) than for unreserved forests (9%). Our analysis holds all other factors constant for these predictions, and we are only evaluating predictions for stands not managed over the growth interval; however, if less management drives greater discrepancy, it is certainly the case that there is less management on reserved forests (by law, there should be none).

### 3.3.5. Ownership

The FVS overpredicts across ownership groups; on national forests and "other public", the overprediction is, respectively, 18% and 15% above predicted FIA values. Overprediction is less for private lands (4%).

### 3.3.6. Tree-Level Models

Tree-level equivalence tests for all trees produced results consistent with what we observed at the stand level. FVS predictions of carbon stocks in stands projected forward ~10 years are equivalent to FIA observations of those stocks within 10% equivalence levels at the means (intercept) and across the range of observations (slope) (Table 4). Annual Growth predictions are equivalent to FIA values at 25% equivalence levels for the mean predictions, but not for the range of predictions agreement tested by the slope. For all the tree and diameter subsets, except for *Sequoia sempervirens* (redwood), annual growth was underpredicted, the tests failed to reject the null hypothesis of dissimilarity and had bootstrap results indicating FVS overstatement of tree growth, given that these failed the null hypothesis below the mean.

**Table 4.** Tree-level equivalence test results for above-ground, non-foliar, tree carbon stocks predicted ~10 years forward and annual carbon growth for subsets of diameter groups and species. Blackened boxes indicate tests that failed to reject the null hypothesis of dissimilarity. The black box with the white asterisk indicates the test where the bootstrapped sample is above the equivalence region.

| Intercept Test | Full | DBH <12.7 | DBH 12.7—25.4 | DBH 25.4—53.34 | DBH 53.34—76.2 | DBH >76.2 | Douglas-fir | Ponderosa Pine | White Fir | Tanoak | Canyon Live Oak | Incense-cedar | Redwood | CA Red Fir | Jeffery Pine |
|---|---|---|---|---|---|---|---|---|---|---|---|---|---|---|---|
| Number of Observations | 69,480 | 5700 | 29,510 | 16,699 | 9512 | 8059 | 10,048 | 4957 | 9058 | 4606 | 4802 | 3778 | 2195 | 3179 | 3292 |
| **Result of Carbon test 10%** | | | | | | | | | | | | | | | |
| Time 2 | | ■ | | | | | | | | | | | | | |
| Annual Growth | ■ | ■ | ■ | ■ | ■ | | | | ■ | ■ | ■ | ■ | * | ■ | ■ |
| **Result of Carbon test 25%** | | | | | | | | | | | | | | | |
| Time 2 | | | | | | | | | | | | | | | |
| Annual Growth | ■ | ■ | ■ | | | | | | | ■ | | ■ | | | |

| Slope Test | Full | DBH <12.7 | DBH 12.7—25.4 | DBH 25.4—53.34 | DBH 53.34—76.2 | DBH >76.2 | Douglas-fir | ponderosa pine | white fir | tanoak | canyon live oak | incense-cedar | redwood | CA red fir | Jeffery pine |
|---|---|---|---|---|---|---|---|---|---|---|---|---|---|---|---|
| **Result of Carbon test 10%** | | | | | | | | | | | | | | | |
| Time 2 | | ■ | ■ | | | | | | | | | | ■ | | |
| Annual Growth | ■ | ■ | ■ | ■ | ■ | ■ | ■ | ■ | ■ | ■ | ■ | ■ | ■ | ■ | |
| **Result of Carbon test 25%** | | | | | | | | | | | | | | | |
| Time 2 | | ■ | | | | | | | | | | | | | |
| Annual Growth | ■ | ■ | ■ | ■ | ■ | ■ | ■ | ■ | ■ | ■ | ■ | ■ | ■ | | |

☐ Rejects Null Hypothesis of Dissimilarity.  ■ Does Not Reject Null Hypothesis of Dissimilarity below Equivalence region.

Exploring the diameter and species subsets, the species subsets are better at producing FIA-observed growth than diameter class subsets. Douglas fir, ponderosa pine, and white fir in the FVS are predicting FIA carbon stocks and growth at the means better than the all-species-combined tree dataset and better than any of the diameter class subsets. Within the diameter subsets, the smallest diameter class, representing trees less than 12.7 cm in diameter, is producing FVS stock and growth predictions that are the most dissimilar to the FIA observations.

## 4. Conclusions

As society increasingly relies on forests to mitigate climate change, notwithstanding their own vulnerability to a changing climate, the need for accurate modeling of changes in forest carbon stocks with time and prospective management becomes ever more pressing. Accuracy improvements begin with rigorous validation of existing models against quality-assured, longitudinal sequences of forest measurements. Ideally, such data consist of a field-collected sample that represents all forest conditions for which modeling projections are needed. We initiated such a validation effort for the FVS covering nearly all the forests in California, a state with outsized forest policy ambitions, and one with a very large and spatially balanced, longitudinal sample of the forests that grow there. We included only the set of stands that did not receive management during the decade-long remeasurement period, a set we dubbed grow-only, because we wanted to have many observations of FIA data that we could assess the FVS against. Moreover, validating FVS predictions for management activities requires knowing exactly when and what kinds of management occurred, data which are not readily available. While our validation does not address the accuracy of FVS projections involving management activity, validation of the base FVS growth model applies to management scenarios involving any growth projection. We find that, at least in California, uncalibrated FVS simulations do not accurately predict growth of carbon or wood volume within a tolerance likely to be considered acceptable for characterizing carbon sequestration. The average growth predicted by the FVS was only equivalent to the FIA measured value at the 25% equivalence level. Even at this level, the slope test failed to reject the hypothesis of dissimilarity, suggesting that the FVS is not predicting growth equivalently to observed growth across the range of observed values. While it is hoped that model calibration might allow the FVS to predict a sufficiently precise growth increment for carbon project evaluation compared to using FVS out-of-the-box, the potential to apply such calibration is contingent on the available inventory data (e.g., either prior measurements of the same trees or radial growth cores of multiple trees per species present), which are not universally available.

FVS predictions of carbon stocks ten years hence, starting from current inventory data, were within 10% of FIA observations. We found that carbon stock predictions were equivalent to observations both at the mean and across stands with different beginning carbon stocks. Using a combination of current inventory and FVS projections may be a valid approach for short term predictions of carbon stocks (up to 10 years). For practical purposes, this suggests that the FVS can appropriately be used, for example, to estimate initial carbon stocks for forests in California, consistent with the California Air Resources Board's (CARB) protocol for forest carbon offsets. Because forest inventory data may predate project initiation by a few years, forest growth models may be used to update inventory data-based carbon stocks data to the project start date. Our findings suggest that for such short periods, stock prediction errors may be within acceptable bounds.

While a 10% equivalence for short-term predictions of average carbon stocks might seem promising for the FVS, the lack of equivalence for growth predictions will compromise the reliability of multi-decadal forest growth projection. For short projections such as the ten years we modeled, future stocks are driven much more by initial stocks than by growth because on average, stocks are large relative to growth. For multi-decade projections, growth will comprise an increasing share of future stocks, a troubling result given the use of the FVS to model growth in carbon stocks under the California Air Resources Board's

(CARB) protocols, which require that management impacts on carbon sequestration be accounted for over ten decades. Given the errors introduced by FVS growth prediction over even a modest ten-year projection, it may be technically infeasible to obtain valid characterizations of management tradeoffs at century timescales. Assigning carbon credits for activities tracked and predicted over shorter periods is rendered problematic by inaccuracy in estimates of carbon growth since the most recent inventory.

Our work illustrates how using multiple methods for validation can produce a better understanding of the nature of model performance. While an essential performance metric, bias is not the entire story. The linear regression-based equivalence tests offer two paths for understanding model performance: the intercept and slope tests. From these tests, we find that the FVS is better at predicting the actual growth observed on FIA plots at the means than it is for predicting accurately across the range of conditions over the forest. This means FVS predictions may be performing better at understanding carbon for an average growing stand, but across the range of stand growth rates, it does not produce results that track reality.

The findings of the tree-level equivalence tests are both consistent with and depart from validation efforts undertaken for other FVS variants. In the Lake States variant, errors were also greatest for the smallest tree diameter classes [25]. These errors predicting the growth of small-diameter trees should raise red flags for applying the FVS where small trees are a substantial component of the forest (e.g., actively managed young growth). For species-specific prediction errors, based on the results of the parametric bootstrap in the equivalence tests, the growth of only one species, *Sequoia sempervirens*, was underpredicted by the FVS. By contrast, Bagdon et al. (2021) found more variability, with some species subsets accurately predicted by FVS and others under- or overpredicted.

Based on our findings, reliable growth projections will almost certainly require calibration of the FVS model. The FVS supports relatively automated calibration if there are previous diameter and height measurements or growth information in the data to be projected. However, many if not most users are working with temporary, not permanent plot data, making this calibration difficult unless radial growth cores were collected. Even with FIA data, there may be too few trees of a species and size class on the plot for calibration to be a viable option. That said, there is little doubt that the FVS, as implemented out-of-the-box, will not generate reliable predictions of growth. We caution users against implementing the FVS in this way and alert them to the need to calibrate. Our findings underscore the need for investment in calibration tools and remeasurement data to support more reliable predictions.

The FVS was designed to support management decisions and was largely fit using data from forests that were actively managed and not in late successional status. Our findings are consistent with the intended use of the FVS, with results most accurate for stands aged less than 50 years and overprediction being greater for less intensively managed reserved forests. However, given that more than half of the forests in California receive essentially no management, and that many if not most carbon projects are linked to benefits assumed to accrue to drastically lengthened rotations and/or withdrawal from management, the dearth of forest growth models capable of accurately predicting prospective carbon growth rates across the full spectrum of forest conditions found in the state is alarming. Inaccurate growth predictions may lead to management regimes, selected for their climate or carbon benefit, that fail to deliver anticipated benefits.

These errors in FVS predictions especially impact national forest lands in California. Public land managers aim to deliver on multiple objectives, including promoting accelerated development of old growth conditions while maintaining high rates of carbon sequestration. However, our results suggest that the FVS overstates growth rates in stands that are older and/or experience less intensive management. Forest plans relying on the base FVS model without stand-specific calibration may misrepresent potential forest growth and associated benefits. Claims of increased carbon sequestration in California

forests, based on out-of-the-box FVS simulations that do not feature active management, deserve particularly tough scrutiny.

These results do not directly measure the potential for the over-crediting of forest carbon offsets. However, our work does apply to landowners and advisors considering deployment of the FVS to evaluate the potential of forest management options to add forest carbon as a revenue stream. Using a grow-only prescription as a baseline against which to compare alternative management scenarios in FVS will, on average, lead to an overprediction of baseline carbon stocks and growth. Significant overpredictions are consistent with work claiming that the FVS introduced over 70% of the modeling uncertainty when modeling increasing rotation length for Improved Forest Management offsets over 100 years [20]. Reliance on the FVS as a basis for evaluating carbon management tradeoffs and expected payouts may lead to disappointment among landowners, carbon credit investors, and enterprises and institutions counting on forests carbon credits to meet climate goals. Notably, we found FVS accuracy is greater on private lands, which are more actively managed, than on public lands, which are less so. This is fortunate, in terms of impacts of FVS accuracy on California's carbon market, because only private lands are currently eligible for participation, not the federal lands on which overstatement of carbon sequestration in the absence of management is most egregious.

Forest carbon protocols and management plans will continue to include options ranging from caretaker management to lengthened rotations as options for landowners to consider, while weighing putative climate mitigation benefits against other forest objectives. Overstated growth rates may enhance the appeal of lengthened rotations and caretaker management for those who seek to manage for carbon benefits—but such benefits may be illusory. Federal forest land contains the overwhelming majority of California's carbon stocks, much of it in stands with high carbon density. The lack of accurate carbon modeling capability for these forests poses a significant challenge for planning climate mitigation responses for these forests. The solution to this problem almost certainly involves better and more transparent processes for carrying out and documenting calibration. Ultimately, this will likely include updating the out-of-the-box model, perhaps with refitted equations based on remeasured FIA plots. Until then, calibration will be essential for any growth projections intended to inform as to the outcome of management alternatives; even then, analysts are advised to consider what is now known as to how specific conditions in the forest being modeled may contribute to FVS growth prediction errors.

**Supplementary Materials:** The following supporting information can be downloaded at https://www.mdpi.com/article/10.3390/f14030604/s1, Figure S1: Diagram of sampling method used in FIA.

**Author Contributions:** Conceptualization, C.H., J.S.F. and V.B.; methodology, C.H., J.S.F. and V.B.; formal analysis, C.H. and J.S.F.; writing—original draft preparation, C.H., J.S.F. and V.B.; writing—review and editing, C.H., J.S.F. and V.B.; visualization, C.H.; project administration, J.S.F. and V.B.; funding acquisition, J.S.F. All authors have read and agreed to the published version of the manuscript.

**Funding:** This research received external funding from the US Forest Service to support this research (Agreements 21-JV-11261979-047 and 16-JV-11261979-116).

**Data Availability Statement:** All data used in this research are publicly available for download. Please contact corresponding author with further inquiries about accessing data.

**Acknowledgments:** We acknowledge the helpful feedback from William C. Stewart and Scott L. Stephens on versions of this manuscript. we are grateful for the dedication of nearly 100 FIA field crew, information management, and analytic staff responsible for collection and quality assurance of the inventory data that supports this validation analysis.

**Conflicts of Interest:** The authors declare no conflict of interest. The funders had no role in the design of the study; in the collection, analyses, or interpretation of data; in the writing of the manuscript; or in the decision to publish the results.

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
