# Peer review of "Validation of Forest Vegetation Simulator Model Finds Overprediction of Carbon Growth in California"

_forests, doi:10.3390/f14030604_

Round 1
Reviewer 1 Report
This manuscript found that the Forest Vegetation Simulator (FVS) overpredicts carbon sequestration in live trees that remain alive 10 ten years later by 27%, on average. The results suggest that forest managers relying on out-of-the-box FVS growth models 17 to forecast carbon sequestration implications of their management of California forests will obtain 18 estimates that overstate the carbon. I think this research is of great application value, especially for the analysis and calculation of carbon sequestration and the construction of models.
I have some minor comments on the content, including:
(1) Line 77: What the “USDA” means?
(2) Lines 181-190: Can the authors give a schematic diagram of the sampling methods, which can be integrated with Figure 1
(3) Line 286-287: I suggest that the variables in all formulas could be represented by letters or abbreviations, and the specific meaning of each letter could be given.
(4) There are many abbreviations in this article. I suggest that if some abbreviations appeared less than 5 times could avoid using them.
(5) Line 445: I suggest not using abbreviations in subtitles or titles.
Author Response
Thank you for the feedback. Please see the attached document.

Reviewer 2 Report
In general the manuscript "Validation of Forest Vegetation Simulator Model Finds Over-prediction of Carbon Growth in California" is very interesting and well-developed, and its contribution to the field of forest ecology is appreciated.
However, some spelling errors were noticed throughout the manuscript that should be addressed before publication. Additionally, distinguishing the discussion and conclusion sections more clearly would help readers better undestand the key findings and implications of the research. Furthermore, it is suggested to consider adding a few more references to the discussion section to enrich the analysis and provide readers with a more comprehensive understanding of the broader context of the research.
Overall, the quality of the work is commented, and the findings are believed to be of great interest to the broader scientific community.
Author Response
Thank you for the feedback. We have addressed spelling errors in the manuscript.
Reviewer 3 Report
Review
The paper by Herbert et al., titled: “Validation of Forest Vegetation Simulator Model finds Overprediction of Carbon Growth in California “, is original work, fitting to the scope of Forests journal.
The paper explored relationship between inventory measurements with empirical model estimates at regional scale. Authors analysed the precision of the model and possible BIAS, which could have significant impact on carbon sequestration estimates in the area. The comparison in the study also differentiates between model variants, site classes, stand age, tree species, DBH classes, reserve status and ownership groups. The presented paper is a practical evaluation of forestry tool widely used in USA.
The paper is overall well composed and easy to read. Abstract is concise and informs the reader about most important findings of the paper. Introduction gives overview of the problematic, but could be improved. Materials and methods are well described, except the initialization and structure of the model. Statistical analyses are adequate. Visual presentation of results is clear and informative. Authors discuss their results extensively and compare their results with other studies.
I recommend minor revision of the paper. Please find my comments below:
Introduction
Authors could add a general introduction to the FVS, is it just a yield table? I know you reference the guidelines, but you could give some overall structural overview. Maybe some graphical visualization of the used FVS modules could be added to introduction?
Materials and methods
Authors do not mention at all what kind of climate they used for the simulation. Was the simulation constrained only by the site index? The initialization of the model is unclear to me.
Line 437: Double dot..
Line 529: How can you simulate and calculate over-production for stands without known age?
Results
Figure 4. Please include all acronyms (CA, NC) in the figure legend.
Discussion
If I understood the model architecture (Crookston 2014, https://doi.org/10.2737/RMRS-GTR-319) and the simulation set-up correctly the authors did not account for the provenances which could significantly impact the tree growth (e.g. Petrik et al. 2022, https://doi.org/10.3390/f14010026). I do not think it is a major issue, but it should be mentioned in the discussion as a limitation of the study.
Author Response
Thank you for the feedback. Please see the responses attached.
